# A Brief Review of Duchenne Muscular Dystrophy Treatment Options, with an Emphasis on Two Novel Strategies

**DOI:** 10.3390/biomedicines11030830

**Published:** 2023-03-09

**Authors:** Ahlke Heydemann, Maria Siemionow

**Affiliations:** 1Department of Physiology and Biophysics, University of Illinois at Chicago, Chicago, IL 60607, USA; 2Center for Cardiovascular Research, University of Illinois at Chicago, Chicago, IL 60607, USA; 3Department of Orthopaedics, University of Illinois at Chicago, Chicago, IL 60607, USA

**Keywords:** Muscular Dystrophy (MD), dystrophin expressing chimeric cells, DEC, systemic-intraosseous administration, muscular dystrophy therapeutics

## Abstract

Despite the full cloning of the Dystrophin cDNA 35 years ago, no effective treatment exists for the Duchenne Muscular Dystrophy (DMD) patients who have a mutation in this gene. Many treatment options have been considered, investigated preclinically and some clinically, but none have circumvented all barriers and effectively treated the disease without burdening the patients with severe side-effects. However, currently, many novel therapies are in the pipelines of research labs and pharmaceutical companies and many of these have progressed to clinical trials. A brief review of these promising therapies is presented, followed by a description of two novel technologies that when utilized together effectively treat the disease in the mdx mouse model. One novel technology is to generate chimeric cells from the patient’s own cells and a normal donor. The other technology is to systemically transplant these cells into the femur via the intraosseous route.

## 1. Introduction

Duchenne Muscular Dystrophy (DMD) is caused by mutations in the dystrophin gene that cause an almost complete lack of the dystrophin protein in the patient. The dystrophin gene is found on the X-chromosome, causing the disease to be X-linked and almost exclusively affecting young boys and young men. Diagnosis usually occurs in the second year of life by the presence of hypertrophic gastrocnemius muscles and the Gower’s maneuver. The maneuver is characterized by the patient’s using his arms to walk up his own body when rising from the floor. The disease goes on to cause weakness in the skeletal, diaphragm and cardiac muscles. Most patients die of cardiac complications and the remainder die from respiratory illnesses in the third or fourth decade of life [1]. Dystrophin mutations that do not completely ablate the protein’s expression or do not alter the protein’s structure severely, cause a milder version of the disease called Becker Muscular Dystrophy (BMD). From the BMD patients with varying levels of dystrophin, we have learned that 20% of normal dystrophin is enough to provide for a mild disease course [2]. This identifies that a low level of dystrophin restoration may alleviate the worst DMD symptoms and can be considered an effective therapy. Despite this low level of dystrophin restoration to be adequate for a remarkably effective treatment, it has not yet been achieved. Therefore, researchers are continuing the effort to bring effective therapies to the patients.

The development of palliative therapies through the years have extended the average lifespan of patients, from approximately 20 years before these palliative therapies to about 40 years with the therapies [1]. These developments include night-time assisted ventilation and cough assistance, anti-heart failure medications and corticosteroids. The steroids aid the skeletal, diaphragm and cardiac muscles, thus keeping the patients out of wheelchairs, and improving respiratory and cardiac function. However, corticosteroids are associated with some severe side effects: weight gain, osteoporosis, mood swings and others. Thus, continued improvements are required.

## 2. Therapeutic Strategies

There are many important advances occurring in the Muscular Dystrophy (MD) research field [3,4]. Many preclinical investigations of therapies are providing a lot of data regarding benefits and possible side effects. These therapies can be separated into seven main groups, based upon the strategy and molecular targets. These strategies are briefly described in Table 1, and more detail is provided below.

### 2.1. In Vivo Gene Correction

The original gene correction strategy was the introduction of the dystrophin gene. It soon became clear that the gene was too large for the standard delivery method—adeno-associated viruses. Therefore, much research has been conducted to design truncated dystrophin DNAs which still retain most of the full-length protein’s functions. Among these are micro- and mini-dystrophin [46]. Scientists are continuing to work on delivering these truncated dystrophins to all diseased muscle groups in the mouse models. For example, a long-term, one dose micro-dystrophin preclinical gene therapy was recently conducted. At 18-months post the single injection, micro-dystrophin was detected by immunostaining and Western blotting in skeletal muscle and heart tissue at levels above normal. Many skeletal muscle and cardiac functional parameters were also improved [47].

Six patients received 6 × 10^11^ or 3 × 10^12^ genomes of mini dystrophin linked to the cytomegalovirus promoter, into one bicep muscle via the AAV2.5 capsid (NCT00428935). Dystrophin DNA was detected via PCR in the injected muscle and, in five/six patients, in the contralateral arm. However, at 90 days post-injection no dystrophin DNA was detected, likely due to an immune response [48,49]. Since this study, additional clinical trials have progressed allowing only patients without pre-existing AAVrh74 antibodies. These trials have been more successful in long term gene expression and phenotype improvements [46].

Many of these MD-treating strategies have been discussed for several years and then a new technique or reagent becomes available to reignite a strategy. This is just what has happened in the in vivo gene correction field [50]. Zinc finger nucleases (ZFN) and transcriptional activator-like effector nucleases (TALENs) have been known since 2005 and 2009 [51,52], respectively, and both demonstrated some promise in pre-clinical studies. However, problems persisted, the most significant being low efficiency of correction, and the issue that still plagues this strategy today—delivery to all cardiac and diaphragm muscle nuclei and to enough skeletal muscle nuclei to produce adequate dystrophin. CRISPR/Cas newly invigorated the field in 2012 [53]. The CRISPR/Cas9 system provides increased site specificity over ZFN and TALENS. In pre-clinical models CRISPR/Cas9 has successfully produced dystrophin expression [54]. High dystrophin expression levels of 60% were also achieved in a dog model of DMD [9]. Continuing experiments with the CRISPR/Cas9 system have identified that single base editing can effectively replace the dangerous double-stranded DNA breaks required of the earlier generations [55,56]. Despite increased site specificity these gene correction techniques still have an issue of off-target mutations. In addition, perhaps a larger issue is that each MD mutation would require a specifically designed gene-correction system. On the highly positive side, in vivo gene correction by ZFN, TALEN or CRISPR have the added potential of being a full cure.

### 2.2. In Vivo mRNA Correction through Read-Through Pharmaceuticals, or Exon Skipping

Read-through techniques take advantage of certain chemicals’ (modified antibiotics) ability to cause ribosomes to ignore (read-through) stop codon sequences during the elongation phase of transcription. PTC124 (Ataluren) is such a chemical that has been tested in the European Union. The ataluren receiving patients had less physical deterioration and delayed respiratory decline, but improvement in the primary outcome—the 6-min walk test (6MWT)—was only observed upon meta-analysis and dividing the patients into three subgroups based upon their baseline 6MWT [10,11]. A long-term (average of 6.3 years of Ataluren treatment) meta-analysis of 11 patients showed an extension in ambulation years and decreases in functional decline in upper limb strength and respiratory parameters [12]. Though these are somewhat promising results, read-through methodology could potentially only help those with non-sense mutations, about 10% of DMD patients [57]. 

Exon-skipping technology also targets the mRNA, in this case as the mRNA is being processed before translation initiates. The exon containing the mutation and, sometimes to maintain reading-frame, flanking exons are targeted to be skipped. Four of these pharmaceuticals Eteplirsen, Drisapersen, Golodirsen and Casimersen are at different stages of clinical trials in the US. Eteplirsen is successful in skipping exon 51. After twenty-four weeks of weekly intra venous dosing of 30 or 50 mg/kg/week, Eteplirsen (NCT02255552) demonstrated increased dystrophin expression, by as much as 52%, and increases in the 6MWT by an average of 67.3 m [15]. Drisapersen (NCT01254019), which is also designed to skip exon 51, has been shown somewhat successful in recently diagnosed young patients, after post hoc analysis. In this trial, 125 patients received 6 mg/kg/week for 48 weeks [17]. All Drisapersen clinical trials have been halted due to lack of effectiveness. In its clinical trial, Golodirsen (which is designed to skip exon 53), was tested in eight patients at 30 mg/kg/week, for 48 weeks, increased dystrophin production and reduced skeletal muscle and respiratory decline were observed (NCT02310906) [18,19]. Continuing issues with exon skipping are that the pharmaceuticals (anti-sense oligonucleotides) are short lived, must be continually administered, can cause toxicity [58], proteinuria [59] and initiate inflammatory signaling cascades [60,61]. Furthermore, exon-skipping strategies only treat a small percentage of patients because there are many MD-causing mutations.

These first two strategies (in vivo-gene or in vivo-mRNA correction) must rely upon a not yet perfected delivery method, such as adeno-associated viruses [62], lipid nanoparticles or a number of other modifications to improve delivery and stability [61,63].

### 2.3. Upregulation of Sarcolemmal Supporting Molecules Such as Utrophin, Integrin-α7 or Sarcospan

Utrophin is a homolog of dystrophin, displaying similar protein structures and functions. In adults, dystrophin is expressed robustly in skeletal and cardiac muscle tissues and minimally in neurons [64,65], while utrophin is expressed at neuro-muscular junctions and in regenerating fibers [66]. Importantly, utrophin can form a stable dystrophin glycoprotein complex because it binds to most of the other complex proteins. Therefore, an early idea was posited that increasing utrophin expression can rescue at least a portion of the DMD phenotype [67,68,69]. This theory has now been proven many times over in the mdx mouse model of DMD, using many different strategies to increase utrophin expression [67]. Although there are still issues, for example, which utrophin-increasing strategy is most promising, and the dosing schedule requires optimized [67]. A small pharmaceutical, Ezutromid, which activates utrophin expression from its native promoter has entered a Phase 1b clinical trial with promising results. With a proper diet, the drug was well-absorbed, achieved desired, functionally useful plasma concentrations and did not incur any serious or severe side effects [22]. However, as often happens when stepping from mouse models to humans, the Phase 2 study (NCT02858362) was halted due to lack of efficacy. It must now be investigated which step was insufficient for clinical improvement; was utrophin upregulation insufficient or did the patients just not respond in the same manner as the mouse models to increased utrophin?

A similar strategy is found in the upregulation of integrin-α7 which functions to attach muscle cells to the basal lamina, the same function (among others) that dystrophin usually serves. By overexpressing integrin-α7 from a transgene, scientists have reduced the pathology of the dystrophin/utrophin double knockout mice [70]. Delivery of integrin-α7 via an adeno-associated virus also benefited the dystrophin/utrophin double knockout mice [71]. To identify patient-applicable strategies, small molecules have been sought to upregulate integrin-α7. One such molecule is SU9516, which is available orally, does increase integrin-α7 and reduces the mdx mouse pathology [23]. Another such molecule is obestatin. Obestatin upregulates utrophin, integrin-α7 and other muscle membrane proteins and decreases mdx pathology [24]. Hopefully clinical trials will soon be initiated.

Sarcospan is a relative newcomer to the world of proteins that can be upregulated to alleviate MD in the mdx mouse [25]. As a newcomer, the methods of Sarcospan over expression have not been as completely investigated, but transgenic overexpression does aid the mice [25]. In addition, sarcospan reduces MD-mediated cardiomyopathy [26].

Upregulation of additional molecules may prove beneficial to the DMD patient. The key to success would be safe and effective delivery of small molecule regulators to DMD patients.

### 2.4. Metabolism Manipulation for DMD Benefit

There are multiple pieces of data indicating that increasing metabolism is helpful for muscles, including muscles of muscular dystrophy patients [72]. Some of these data stems from early observations that slow fibers are somewhat resistant to dystrophic pathologies [73]. Further evidence then came from positive results of switching metabolic signaling through the upregulation of pAMPK [74]. There are multiple strategies to upregulate pAMPK [74,75,76]. Metformin alters cellular metabolism by reducing the efficiency of the electron transport chain and thus ‘tricking’ the cell into thinking it is low on ATP. The cell responds by increasing mitochondrial production and metabolism [77]. This increased metabolism rate is likely beneficial, but the associated signaling cascades (pAMPK and PGC1α) of a starved cell also benefit the muscle cells [78]. The most protective metabolic manipulations in mouse models are not yet decided upon, but human trials of metformin are underway. The most significant aspect of treating MD with metformin is that metformin is already FDA approved and has few lasting side-effects [77]. Metformin is the primary medication for type 2 diabetes, is now also often prescribed for clinically necessary weight loss and has proven safe for over 20 million patients in 2020 in the USA (https://clincalc.com/drugstats/drugs/metformin (accessed on 13 January 2023)). When metformin was co-administered with L-arginine to DMD patients, the therapy was well tolerated and changes in the mitochondrial metabolic pathways and nitric oxide signaling cascades were changed to a possibly more favorable position. However, no statistically significant clinical benefits were observed, likely only because the study was underpowered (NCT01995032) [27,28]. 

PGC1α upregulation has been identified as a central signaling nexus for mitochondrial and skeletal muscle health [79,80]. Resveratrol, an extract of dark grapes, can upregulate PGC1α with minimal side-effects. In addition, resveratrol decreases reactive oxygen species, increases oxidative capacity, increases mitochondrial biogenesis and upregulates utrophin. In many mdx experiments, resveratrol proved successful at reducing pathology, reviewed in [81]. A clinical trial (NCT01856868) increasing PGC1α with epicatechin (a structural homologue of a family of PGC1-alpha activating steroid hormones) produced some benefits for the patients, such as increases in elbow flexion and increases in utrophin. 

### 2.5. Designing Novel Steroids Such as Deflazacort and Volmerone

These advanced generation steroids are designed to provide the benefits, but not the side effects of traditional steroids [31]. Deflazacort has proven to be more effective than prednisone and with decreased side effects. The deflazacort patients required wheelchairs later in life and fewer patients developed scoliosis and it developed at a later age [30]. Volmerone has also proven as effective as deflazacort and has lower side effects than prednisone (NCT02760264, 02760277, 03038399) [32]. A further area of research is the effort to identify the best dosing regimens. In mice, daily or weekly administration of 5mg/ml prednisone demonstrated significant benefits, while only the daily regimen elicited side effects [82]. A portion of the benefit of weekly dosing regimens may be the realignment of the synchronicity usually observed in acute muscle injuries in non-DMD animals and patients. 

### 2.6. Repurposing of Pharmaceuticals That Are Designed for Use against Other Diseases Has Many Advantages

This strategy saves time and money usually devoted to research, development, pharmacokinetics and safety studies. Saving this money will help to fight any disease but are more important when combatting a rare disease such as MD [83]. Read-through antibiotics, metformin and next-generation antibiotics are examples of this strategy that have already been discussed. 

Tamoxifen is usually prescribed to combat breast cancer and acts by competing for the estrogen receptor [84]. In the pre-clinical MD arena, long term tamoxifen can reduce fibrosis, increase force, aid in regeneration and cause myogenesis to decrease MD pathology [85,86,87]. In a 3-year clinical trial (NCT02835079), patients showed maintenance of muscle and respiratory functions and decreased creatine kinase levels [33].

A common low-density lipoprotein-lowering pharmaceutical—simvastatin—has proven to benefit the mdx mouse model of DMD. Simvastatin reduces fibrosis, reactive oxygen species and inflammation in the mouse model, benefitting both skeletal and cardiac muscle function [34,35]. Clinical trials have not yet been initiated.

Zidovudine (AZT) inhibits purinergic receptor (P2RX7) signaling, was developed in the fight against HIV/Aids and in a mouse-model assists against MD by the same molecular cascade [36]. Following two weeks of treatment, the mice had less inflammation, fibrosis, membrane damage, increased grip strength and distance in a test run [36]. Clinical trials have not yet been initiated.

### 2.7. Cell Transplantation Investigations

Cell transplantation has long been seen as a potential therapy for DMD [88] and was recently reviewed [89,90,91,92]. Issues specific to this therapy are identifying the best muscle regenerating cell type, immune rejection of the donated cells, immune rejection of dystrophin and insufficient engraftment. Some of these transplant experiments have reached the state of clinical trial. 

For example, a Phase I/II study involved the introduction by “high-density injections” of thirty-million donor myoblasts into the extensor carpi radialis of the forearm (NCT02196467), while the contralateral forearm received saline. The donor was one of the patient’s parents and the recipient was treated with tacrolimus for immunosuppression. Eight of nine patients expressed dystrophin (between 3.5 to 26% of normal) by immunofluorescence at 4-weeks post-injections [37]. Another trail introduced cardiosphere-derived cells into coronaries of 13 patients with substantial myocardial fibrosis (NCT 02485938). Importantly, no immune suppression was used. Cardiac parameters were measured by MRI after 12 months. The treated patients had significantly reduced scar size and improvement in inferior wall systolic thickening. The treated patients also had improvements in some of their upper limb tests [38].

## 3. Future of Cell-Based Therapeutics

### 3.1. Choosing the Correct Cell Type

Several cell types have been investigated for their abilities to rescue DMD in the mdx mouse model. Initially, scientists used neonatal mouse myoblasts, additional studies utilized satellite cells (muscle resident stem cells), induced pluripotent stem cells (iPSCs), normal adult myoblasts, side population cells, bone marrow cells, pericytes, mesenchymal stem cells (MSC) and mesoangioblasts [90,92]. Many of these cells (especially the MSCs) have the added benefit of secreting cytokines that are myotrophic and/or myogenic [93]. At first glance, satellite cells would appear to be the best candidates to repopulate injured muscles. However, satellite cells have issues; very few can be isolated, propagation in culture reduces their stem-ness, they self-aggregate by blocking small vessels and never achieve muscle localization, even when injected directly into the muscle few cells survive [90,92].

There is also the decision between autologous or allogenic cells. Autologous procedures utilize the patient’s own cells, usually genetically corrected ex vivo, expanded and then reintroduced into the patient. These cells have the advantage of not requiring immune suppressive pharmaceuticals. Allogenic cells are from a different person, a true donor. These cells have the advantage that they contain normal dystrophin, so they only need to be expanded ex vivo. However, unless the cells are from a perfectly matched donor the patients will require immune suppressive medications for the remainder of their lives.

### 3.2. Production of Novel Chimeric Cells Which Avoids Immune Rejection

Many of the MD transplant therapies would require donor matching or continuous immunosuppressants. Donor matching requires identical blood types and identical Human Leukocyte Antigen (HLA) between the healthy, normal donor and the recipient. This test must be conducted to match each patient. Currently more than 1200 HLA variants have been identified, making matching difficult, time-consuming and expensive [94]. Close relatives are more likely to be matches, making the statistics better in some situations. The recipient’s blood must also be checked for anti-HLA antibodies possibly formed from a previous transplantation or blood transfusion. Additional tests are performed to identify harmful interactions between donor and host. Despite these many tests, graft-versus-host-disease still occurs in more than 25% of transplant recipients [95].

If a perfectly matched donor cannot be found, immunosuppressant pharmaceuticals are an option. These drugs are likely to be needed for the life of the patient and missing a single dose can cause severe reactions. These drugs are also associated with some severe side-effects. The patients will have increased chances of infections and may have coughs and colds that require more time to dissipate. Therefore, additional research is required to successfully bring cell transplants to the clinic.

One method of avoiding the immune rejection dilemma is to mark the transplanted organs, tissues or cells as self. An interesting method to mark donor cells as self is to fuse the donor cells to the recipient’s cells in ex vivo cell culture, to create Dystrophin Expressing Chimeric cells (DEC). To ensure chimerism, the cells are first labelled with red or green dyes, the yellow/orange fused cells are fluorescently sorted and expanded in culture. The sorting step is very important as some donor cells will fuse with other donor cells; if enough of these cells are transplanted an immune reaction would occur. In the desired yellow/orange chimeric cells, the donor cell and nucleus provide the dystrophin while the recipient cell and nucleus provide the cell surface proteins that mark the fused cell as self.

This fusion chimerism has been proven to induce tolerance in vascularized composite allotransplantation and bone marrow transplantation studies [96,97,98]. 

### 3.3. The DEC Fusion Techniques for Muscular Dystrophy Have Been Perfected In Vitro and Preclinically in the mdx Mouse

The proof-of-principle chimeric cell experiments demonstrated that myoblast cells from two different mice could be successfully fused without losing muscle stemness and without losing the ability to proliferate. Furthermore, when myoblasts from an mdx mouse were fused with myoblasts from a wildtype mouse: MB^N^/MB^mdx^ the chimeric cells expressed dystrophin in vitro. After sorting for and expansion of the fused cells, the MB^N^/MB^mdx^ cells were injected directly into the gastrocnemius muscle (GM) of mdx mice. Upon analysis at 90-days post-transplant, dystrophin was found to be expressed and localized correctly to the sarcolemma. The injected GM samples also demonstrated increased strength by in vivo and ex vivo muscle tests and increased resistance to fatigue [40,41].

The next step was to analyze the possibility that normal human myoblast cells could be fused to DMD patient myoblasts and still retain their ability to differentiate into muscle cells and proliferate [40]. After in vitro confirmation of fusion, differentiation and proliferation, these human fused cells were injected into the GM of mdx/scid mice and followed for 90 days. Again, the DEC cells repopulated the GM and expressed dystrophin and restored the expression of other dystrophin glycoprotein complex members. Furthermore, after 90 days the DEC-injected GM produced significantly more force [40]. 

An additional—often raised—cell transplant question is which cells will proliferate best and repopulate the muscle best. Based upon many previous publications, the authors also tested the fusibility of murine myoblasts with mesenchymal stem cells (MSC). The fused MB^N^/MSC^N^ DEC cells performed well in the in vivo battery of tests. They fused well, maintained their muscle stem-ness, expressed dystrophin, and proliferated well. These cells also passed an in vitro Comet Assay, which investigates possible genotoxicity. None of the MB^N^, MSC^N^ or MB^N^/MSC^N^ DEC cells displayed any genotoxicity. Furthermore, to evaluate any possible immune reactions elicited by the transplanted chimeric cells the Mixed Lymphocyte Reaction was utilized. Again, the three chimeric cell types passed by displaying reduced alloreactivity. These DEC cells also passed the localized GM transplantation in vivo tests. At 90 days post-transplantation, dystrophin expression was present and functional tests demonstrated improvement over vehicle-injected controls [45]. 

### 3.4. Choosing the Correct Delivery Route

The chimeric cell technique overcomes many of the issues that surrounded cell transplantation experiments. The chimeras evade the immune response thus allowing efficient engraftment and proliferation when locally injected. As important as ensuring that the cells are right is the use of the correct delivery route. Scientists have attempted intravenous in the tail vein [99] or intraarterial transplantation [100]. None of these produced adequate engraftment or dystrophin expression. Very few of the precious cells implanted into the muscles. A newly tested delivery route is through intraosseous delivery (IO). This route has historically been used in an emergency setting when a vein cannot be accessed. This route was now to be tested for systemic DEC cell delivery [43,44]. These experiments could then also analyze the global effectiveness of the DEC cells in alleviating cardiac and respiratory pathologies. 

Intraosseous delivery has now been effectively utilized preclinically to treat the mdx mouse model of DMD. Human DEC cells (MB^N^/MB^DMD^) were identified by immunofluorescence to have localized to the gastrocnemius, diaphragm, and cardiac muscles at least to 90 days post transplant [43]. The expression patterns correlated with improved, quantitative histopathology and function in all three muscle tissues. The quantitative histology measurements included the analysis of inflammatory foci. As inflammation is ubiquitous with DMD, the found decrease in this characteristic also indicates benefits from the DEC transplantation. Quantitative histology measurements also included fibrosis analysis, and again, the DEC-treated animals displayed reduced fibrosis. The functional improvements include force generation capabilities of the GM, improved ejection fraction and fractional shortening for the heart, and improved tidal volume and relaxation times for the diaphragm (improving respiratory function). These same benefits (in GM, diaphragm and cardiac muscles) were observed in mice that had received DEC therapy via the intraosseous route after 180 days [39,42]; thus, demonstrating that the DEC cells also have longevity after their engraftment. 

Importantly, the mice were assessed for additional safety measures at 180 days after intraosseous transplantation of human DEC cells (MB^N^/MB^DMD^). By flow cytometry, there was negligible presence of DEC cells in the off-target organs (blood, bone marrow, lung, liver, and spleen) and a dose-dependent increase in DEC cells in the target organs (heart, diaphragm, and gastrocnemius). Furthermore, magnetic resonance imagining was utilized to demonstrate lack of tumors. The lack of tumors was verified by a trained veterinarian conducting a complete necropsy at the study’s endpoint [39].

## 4. Conclusions

Despite the many years of corticosteroids being the sole pharmaceutical prescribed for DMD patients, many promising therapies are on the horizon for these patients. The most important point is the many different strategies being investigated. This will make success of at least one of the pharmaceuticals more likely. A further benefit to these multiple strategies is that one can envision combination therapies being formulated. These combination therapies may allow lower doses of the individual components, reducing the side-effect dilemma, but still—in combination—producing benefits. Additional therapies—not discussed in this review—that could be combined with the listed therapies include specific exercise regimes [101] and nutritional modifications [72]. With these many avenues and momentum, it is not surprising that many clinical trials to combat DMD are occurring [3,102].

There are several barriers that MD therapies must overcome. The main barriers are delivering therapy to all muscle groups, effectively treating the disease in skeletal, diaphragm and cardiac muscles once delivered, sustaining treatment, and limiting severe side effects. Among the therapies currently being investigated is cell transplantation. As with other transplantation scenarios, immune rejection is an additional barrier. There are many different cell types, different pre-transplant manipulations of the cells and different routes of delivery. A relatively new strategy is the making of Dystrophin Expressing Chimeric cells (DEC cells), for transplantation. These cells result from the fusion of the patient’s muscle cells with those of a healthy donor. The recipient’s cells provide the ‘self’ surface markers, thereby avoiding any immune rejection of the transplant; meanwhile, the cells from the healthy donor provide full-length dystrophin. To overcome the other barriers, intraosseous delivery has been used to deliver the cells systemically. Once delivered, the cells specifically target the muscle tissues and effectively treat the disease in canonical skeletal, diaphragm and heart muscles and maintain treatment for at least 180 days. DEC therapy is not dependent on the genetic mutation of the DMD patient, thus making DEC a universal therapy for all DMD patients. It is also possible that the combination of DEC cells with intraosseous delivery can benefit other muscular dystrophies. In addition, as DEC therapy does not require viral vectors, it can be readministered. The therapy is not associated with any genetic manipulation and therefore involves no risk of off-target mutations. Having proven itself pre-clinically, this therapy has now entered a Dystrogen-administered clinical trial in Poland, approved by the Poznan University Hospital Bioethics Committee (approval number 46/2019).

## Figures and Tables

**Table 1 biomedicines-11-00830-t001:** Highlights of Promising Therapeutic Strategies.

Strategy	Pros	Cons	Specific Therapy	Pre-Clinical/Clinical Results	Refs
1. In vivo gene correction	If stem cells are repaired this may be a true cure. Mutation independent	May cause off-site mutations.Safe delivery to all muscle cells is not yet perfected.	SRP-9001. rAAV-rh74 delivery of micro-dystrophin, 1 injection.	NCT03375164, 03769116, 04626674. Patients had long-term gene expression and phenotype improvements. Phase 3 is now recruiting. Some patients developed antibodies to the virus.	[5,6]
			Pf-06939926. rAAV9 delivery of mini-dystrophin, 1 injection, 2 doses.	NCT0336502, NCT04281485, NCT05429372. Patients had gene expression, and an average 3.5-point increase in the NSAA score. A total of 40% of patients experienced vomiting and/or nausea. Phase 3 trial on hold due to a patient’s death.	[7]
			SGT-001, rAAV9 delivery of micro-dystrophin, 1 injection, 2 doses.	NCT03368742. Variable dystrophin expression. Phenotype improvement in 6MWT and NSAA scores. Many severe adverse effects; liver and kidney injuries.	[8]
		Mutation specific	CRISPR/Cas9	Achieved 60% of normal dystrophin in a canine DMD model	[9]
2. In vivo mRNA correction	These have demonstrated clinical benefits.	Safe delivery to all muscle cells is not yet perfected. Some adverse drug reactions. Must be continually re-administered. Mutation specific	Read through; Ataluren	NCT01826487, NCT01557400. Reduces many of the disease symptoms, such as loss of ambulation and respiratory decline	[10,11,12]
			Exon 51 skipping; Eteplirsen	NCT02255552. Small, if any, improvements over the control group at 96 weeks post treatment. Delay in pulmonary decline.	[13,14,15]
			Exon 53 skipping; Vitolarsen	Achieved an average of 5.9% of normal dystrophin levels after 20 weeks of treatment	[16]
			Exon 51 skipping; Drisapersen	NCT01254019. Some benefit with post hoc statistics in the 6MWT, clinical trials terminated	[17]
			Exon 53 skipping; Golodirsen	NCT02310906. Decreased muscle function decline.	[18,19]
			Exon 45 skipping; Casimersen	NCT02500381. Confirmed safety.	[20,21]
3. Upregulation of supporting molecules	Will treat most DMD patients. Low side-effects.		Utrophin	NCT02858362. Study was halted due to lack of efficacy.	[22]
			Integrin-α7; SU9516	PC., Slows disease progression	[23]
			Integrin-α7; Obestatin	PC. Increased force production and other aspects of the mdx phenotype	[24]
			Sarcospan	PC, decreases mdx muscle pathology including cardiomyopathy	[25,26]
4. Enhancing muscle metabolism	FDA-approved for Type 2 diabetes		Increase pAMPK; Metformin	NCT01995032. No DMD reducing results.	[27,28]
			Increase PGC1α	NCT01856868. Some benefit for the patients.	[29]
5. Novel steroids	Fewer side-effects	May still decrease patient’s immune response	Deflazacort	A retrospective patient study identified benefits of deflazacort over prednisone.	[30]
			Vamorolone	PC. Vamorolone reduces fibrosis, inflammation and cardiomyopathy in mdx mice with reduced side effects. NCT02760264, 02760277, 03038399. Improvement in muscle function over natural history values and fewer side-effects then with corticosteroids.	[31,32]
6. Repurposing pharmaceuticals	Less expensive. Already passed human safety trials.		Tamoxifen	NCT02835079. Lower decreases in muscle and respiratory functions.	[33]
			Simvastatin	PC. Reduced pathology and increased muscle function.	[34,35]
			Zidovudine (AZT)	PC. Reduced pathology and increased muscle function.	[36]
7. Cell Transplants		Requires immune suppression.	Myoblasts	NCT02196467. Local high-density cell injections with immune suppression. Dystrophin was detected at the injection site at 4-weeks post-injections.	[37]
	Additionally benefitted skeletal muscles.		Cardiospheres	NCT02485938. Coronary injections without immune suppression. At 12-months post treatment only the treated patients had reduced size of myocardial scars.	[38]
	No immune suppression is needed. Intraosseous, systemic delivery		Dystrophin expressing chimeric cells (DEC)	PC. Skeletal, cardiac and diaphragm muscle improvements up to 180 days post single injection.	[39,40,41,42,43,44,45]

NCT: NIH clinical trials number, this number indicates clinical trial data on DMD patients will follow; PC: denotes preclinical data on the mdx mouse; 6MWT: 6-min walk test; NSAA: North Star ambulatory assessments.

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
