# Peer review of "A Brief Review of Duchenne Muscular Dystrophy Treatment Options, with an Emphasis on Two Novel Strategies"

_biomedicines, 2023, doi:10.3390/biomedicines11030830_

Round 1

Reviewer 1 Report

The manuscript entitled “Two novel techniques successfully treat the Duchenne muscular dystrophy mouse model” by Heydemann & Siemionow revised the most successful and state of art therapies developed, tested, and applied in pre-clinical and in clinical trials used for Duchenne muscular dystrophy. The manuscript briefly described most of the pre-clinical strategies and clinical therapies developed for DMD aiming to describe the technique and results that the Authors are experts, the dystrophin expressing chimeric (DEC) cells transplanted into the bone marrow of mdx mouse model of DMD.  The manuscript is clear, well-structured, and organized bringing to the reader the current state of the different approaches and therapies for DMD.

The major comment to the review is the superficiality given by the Authors for most pre-clinical and clinical therapies. The Authors should include more information regarding the composition, dosages, time/ duration, and a better description of the results obtained. The current status does not provide the reader with a novel and improvement of information in comparison to others reviews recently published.

There are minor aspects that the Authors should address to improve the quality of the manuscript, and thus a contribution to the researchers, family, and patients as is described below.

1)      The title is open, incomplete, and vague, and does not represent the content of the manuscript. The Authors should modify the title accordingly to the content and aim of the manuscript.

2)      The Abstract seems incomplete and impartial for a review. The authors should also mention other topics addressed in the manuscript and give a direction to the reader.

3)      The Authors should consider adding references for the findings described in lines 17-25.

4)      The Authors should improve the writing of the manuscript in several passages, where one can find a lack of polishing and maturation of the idea. For instance, in lines 37-39, the word “corticosteroids” is repeated three times, which can be done once for better understanding. Another example is seen on line 123, where the sentence has only five words missing the entire context and thus the fluidity of the idea.

5)      Table 1 is unbalanced in terms of the description of each strategy; Although points 1 and 2 are similar, from point 3 on, the Authors superficially described the main findings, limiting sometimes only to describe the negative points of the research. Moreover, there is a missing citation on point 1 and there is no number 6, jumping from 5 to 7, creating an inconsistency in the main text when describing the table information in the main text.

6)      Line 116: The references in the sub-title should be removed and add a proper sub-title for the chapter.

7)      Line 118: The developmental expression that differentiates utrophin and dystrophin should be explained.

8)      Lines 151-159: The Authors should correct the text and add a sub-title for point 2.4

9)      Reference 11, 15, and 26: Please incorporate the correct citation. There are papers available for these NCTs.

10)   Reference 36: This reference cannot be found. Please replace it with a published paper or remove it.

Author Response

Response to reviewers, biomedicines-2210931

Thank you for your hard work making this manuscript more complete and easier to read.

We have addressed all of the reviewer’s concerns, as listed below and seen in the manuscript edits.

Reviewer #1

The manuscript entitled “Two novel techniques successfully treat the Duchenne muscular dystrophy mouse model” by Heydemann & Siemionow revised the most successful and state of art therapies developed, tested, and applied in pre-clinical and in clinical trials used for Duchenne muscular dystrophy. The manuscript briefly described most of the pre-clinical strategies and clinical therapies developed for DMD aiming to describe the technique and results that the Authors are experts, the dystrophin expressing chimeric (DEC) cells transplanted into the bone marrow of mdx mouse model of DMD.  The manuscript is clear, well-structured, and organized bringing to the reader the current state of the different approaches and therapies for DMD.

The major comment to the review is the superficiality given by the Authors for most pre-clinical and clinical therapies. The Authors should include more information regarding the composition, dosages, time/ duration, and a better description of the results obtained. The current status does not provide the reader with a novel and improvement of information in comparison to others reviews recently published.

We have made numerous additions to the manuscript to completely review the field of preclinical and clinical trials.

There are minor aspects that the Authors should address to improve the quality of the manuscript, and thus a contribution to the researchers, family, and patients as is described below.

1)      The title is open, incomplete, and vague, and does not represent the content of the manuscript. The Authors should modify the title accordingly to the content and aim of the manuscript. We have changed the title to reflect the composition of the manuscript.

2)      The Abstract seems incomplete and impartial for a review. The authors should also mention other topics addressed in the manuscript and give a direction to the reader. The new abstract reflects the reviews subject matter.

3)      The Authors should consider adding references for the findings described in lines 17-25. We have now done this.

4)      The Authors should improve the writing of the manuscript in several passages, where one can find a lack of polishing and maturation of the idea. For instance, in lines 37-39, the word “corticosteroids” is repeated three times, which can be done once for better understanding. Another example is seen on line 123, where the sentence has only five words missing the entire context and thus the fluidity of the idea. Thank you for identifying our careless proof reading, we have reread the manuscript and made numerous changes including those specifically suggested by you.

5)      Table 1 is unbalanced in terms of the description of each strategy; Although points 1 and 2 are similar, from point 3 on, the Authors superficially described the main findings, limiting sometimes only to describe the negative points of the research. Moreover, there is a missing citation on point 1 and there is no number 6, jumping from 5 to 7, creating an inconsistency in the main text when describing the table information in the main text. We have bolstered the table points 3-7 and have corrected the numbering.

6)      Line 116: The references in the sub-title should be removed and add a proper sub-title for the chapter. This has been done.

7)      Line 118: The developmental expression that differentiates utrophin and dystrophin should be explained. This has been done.

8)      Lines 151-159: The Authors should correct the text and add a sub-title for point 2.4. This has been done.

9)      Reference 11, 15, and 26: Please incorporate the correct citation. There are papers available for these NCTs. The clinical trials references have been replaced by the proper manuscript references. We could not find the appropriate manuscripts for the Pfizer compound (pf-06939926) and the Serepta compound (SGT-001) so web sites were included.

10)   Reference 36: This reference cannot be found. Please replace it with a published paper or remove it. This has been done.

Reviewer 2 Report

Summary

This manuscript reviews the therapeutic progress for Duchenne muscular dystrophy (DMD). In the first half, the authors briefly overviewed the therapeutic strategies for DMD such as gene correction, pharmaceutical agents, and cell transplantation. In the last half, the manuscript focused on the novel developed technologies for generation and delivery of the cells for DMD treatment. This reviews compactly introduces the recent progresses and knowledges of the DMD studies, providing useful information in this field. Some minor points below need to be revised before publication.

Minor points

1.          Lines 45-46: “These therapies can be separated into 8 main groups...” but the Table contains only seven groups (lacking No. 6). And the section numbers of 2.6 and 2.7 are not corresponding to the Table.

2.          Line 47: “Table 1” may be “Table” because the manuscript has only one table. And the title of the Table should be given.

3.          Line 116: The title of the section 2.3 should be provided.

4.          Line 151: The title of the section 2.4 should be provided.

Author Response

Response to reviewers, biomedicines-2210931

Thank you for your hard work making this manuscript more complete and easier to read.

We have addressed all of the reviewer’s concerns, as listed below and seen in the manuscript edits.

Reviewer #2

This manuscript reviews the therapeutic progress for Duchenne muscular dystrophy (DMD). In the first half, the authors briefly overviewed the therapeutic strategies for DMD such as gene correction, pharmaceutical agents, and cell transplantation. In the last half, the manuscript focused on the novel developed technologies for generation and delivery of the cells for DMD treatment. This reviews compactly introduces the recent progresses and knowledges of the DMD studies, providing useful information in this field. Some minor points below need to be revised before publication.

Minor points

  1. Lines 45-46: “These therapies can be separated into 8 main groups...” but the Table contains only seven groups (lacking No. 6). And the section numbers of 2.6 and 2.7 are not corresponding to the Table. These items have been corrected.
  2. Line 47: “Table 1” may be “Table” because the manuscript has only one table. And the title of the Table should be given. These items have been corrected.

  1. Line 116: The title of the section 2.3 should be provided. This has been done.
  2. Line 151: The title of the section 2.4 should be provided. This has been done.

Round 2

Reviewer 1 Report

The Authors of the manuscript now entitled "A current review of preclinical and clinical trials to treat Duchenne muscular dystrophy" addressed most of the comments suggested in the first revision. 

The only minor comment on the revised form of the manuscript is regarding the abstract. Although the Authors changed the abstract, there is no improvement regarding in the quality of describing the content and aim of the review. 

Author Response

Dear Editor and Reviewers,

Thank you for rereading the manuscript: A current review of preclinical and clinical trials to treat Duchenne muscular dystrophy, biomedicines-2210931. We have reworked the abstract to reflect the manuscript’s content more closely.

Sincerely,

Ahlke Heydemann
